# Towards Optimal Strategies for Training Self-Driving Perception Models in Simulation

**David Acuna**,*  **Jonah Philion**,*  **Sanja Fidler**
NVIDIA,  University of Toronto,  Vector Institute
{dacunamarrer, jphilion, sfidler}@nvidia.com

## Abstract

Autonomous driving relies on a huge volume of real-world data to be labeled to high precision. Alternative solutions seek to exploit driving simulators that can generate large amounts of labeled data with a plethora of content variations. However, the domain gap between the synthetic and real data remains, raising the following important question: *What are the best way to utilize a self-driving simulator for perception tasks?* In this work, we build on top of recent advances in domain-adaptation theory, and from this perspective, propose ways to minimize the reality gap. We primarily focus on the use of labels in the synthetic domain alone. Our approach introduces both a principled way to learn neural-invariant representations and a *theoretically inspired* view on how to sample the data from the simulator. Our method is easy to implement in practice as it is agnostic of the network architecture and the choice of the simulator. We showcase our approach on the bird's-eye-view vehicle segmentation task with multi-sensor data (cameras, lidar) using an open-source simulator (CARLA), and evaluate the entire framework on a real-world dataset (nuScenes). Last but not least, we show what types of variations (e.g. weather conditions, number of assets, map design and color diversity) matter to perception networks when trained with driving simulators, and which ones can be compensated for with our domain adaptation technique.

## 1   Introduction

The dominant strategy in self-driving for training perception models is to deploy cars that collect massive amounts of real-world data, hire a large pool of annotators to label it and then train the models on that data using supervised learning. Although this approach is likely to succeed asymptotically, the financial cost scales with the amount of data being collected and labeled. Furthermore, changing sensors may require redoing the effort to a large extent. Some tasks such as labeling ambiguous far away or occluded objects may be hard or even impossible for humans.

In comparison, sampling data from self-driving simulators such as [46, 2, 6, 13] has several benefits. First, one has control over the content inside a simulator which makes all self-driving scenarios equally efficient to generate, indepedent of how rare the event might be in the real world. Second, full world state information is known in a simulator, allowing one to synthesize *perfect labels* for information that humans might annotate noisily, as well as labels for fully occluded objects that would be impossible to label outside of simulation. Finally, one has control over sensor extrinsics and intrinsics in a simulator, allowing one to collect data for a fixed scenario under any chosen sensor rig.

Given these features of simulation, it would be tremendously valuable to be able to train deployable perception models on data obtained from a self-driving simulation. There are, however, two critical challenges in using a driving simulator out of the box. First, sensor models in simulation do not mimic real world sensors perfectly; even with exact calibration of extrinsics and intrinsics, it is extremely

---

*denotes equal contribution, https://nv-tlabs.github.io/simulation-strategies

35th Conference on Neural Information Processing Systems (NeurIPS 2021).

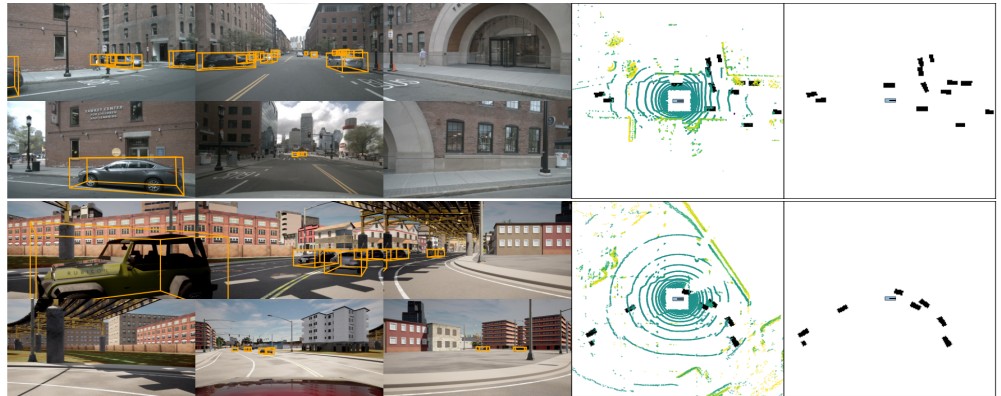

Figure 1: **nuScenes vs CARLA** Visualization of real-world data & labels from nuScenes [7] (top) and synthetic data & labels we obtain from CARLA [13] (bottom). As in nuScenes, we scrape 6 cameras, 1 LiDAR, ego-motion (for temporal fusion of LiDAR scans), and 3D bounding boxes from CARLA. On the right, we show the target binary image for the bird's-eye-view vehicle segmentation task that we consider in this paper [43, 37, 22].

hard to perfectly label all scene materials and model complicated interactions between sensors and objects – at least today [34, 8, 56, 42]. Second, real world content is difficult to incorporate into simulation; world layout may be different, object assets may not be diverse enough, and synthetic materials, textures and weather may all not align well with their real world counterparts [51, 6, 19, 55]. To make matters worse, machine learning models and more specifically neural networks are very sensitive to these discrepancies, and are known to generalize poorly from virtual (source domain) to real world (target domain) [52, 14, 44].

**Contributions and Outline.** In this paper, we seek to find the best strategies for exploiting a driving simulator, that can alleviate some of the aforementioned problems. Specifically, we build on recent advances in domain adaptation theory, and from this perspective, propose a theoretically inspired method for synthetic data generation. We also propose a novel domain adaptation method that extends the recently proposed $f$-DAL [3] to dense prediction tasks and combines it with pseudo-labels.

We start by formalizing our problem from a DA perspective. In Section 2, we introduce the mathematical tools, generalization bounds and highlight main similarities and *differences* between standard DA and learning from a simulator. Our analysis leads to a simple but effective method for sampling data from the self-driving simulator introduced in Section 3. This technique is simulator agnostic, simple to implement in practice and builds on the principle of reducing the distance between the labels' marginals, thereby reducing the domain-gap and allowing adversarial frameworks to learn meaningfully in a representation space. Section 4 generalizes recently proposed *discrepancy* minimization adversarial methods using Pearson $\chi^2$ [3] to dense prediction tasks. It also shows a novel way to combine domain adversarial methods with pseudo-labels. In Section 6, we experimentally validate the efficacy of our method and show the same approach can be simply applied to different sensors and data modalities (oftentimes implying different architectures). Although we focus on the scenario where labeled data is not available in the target domain, we show our framework also provides gains when labels in the target domain are available. Finally, we show that variations such as number of vehicle assets, map design and camera post-processing effects can be compensated for with our method, thus showing what variations are less important to include in the simulator.

## 2   Learning from a Simulator: A Domain Adaptation Perspective

We can learn a great deal about a model's generalization capabilities under distribution shifts by analyzing its corresponding binary classifier. Therefore, building on the existing work in domain-adaptation theory [5, 4, 59, 3], we assume the output domain be $\mathcal{Y} = \{0, 1\}$ and restrict the mathematical analysis to the binary classification setting. Section 6 shows experiments in more general settings, validating the usefulness of this perspective and our algorithmic solution which is inspired by the theoretical analysis presented in this section. Specifically, here, we interpret learning from a simulator as a domain adaptation problem. We first formulate the problem and introduce the notation that will be used throughout the work. Then, we introduce mathematical tools and generalization bounds that this work builds upon. We also highlight main similarities and *differences* between standard DA and learning from a simulator.

We start by assuming the self-driving simulator can automatically produce labels for the task in hand (as is typical), and refer to data samples obtained from the simulator as the synthetic dataset (S) with *labeled* datapoints $S = \{(x_i^s, y_i^s)\}_{i=1}^{n_s}$. Let's also assume we have access to another dataset (T) with *unlabeled* examples $T = \{(x_i^t)\}_{i=1}^{n_t}$ that are collected in the real world. We emphasize that *no labels are available* for T. Our goal then is to learn a model (hypothesis) $h$ for the task using data from the labeled dataset S such that $h$ performs well in the real-world. Intuitively, one should incorporate the unlabeled dataset T during the learning process as it captures the real world data distribution.

Certainly, there are different ways in which the dataset S can be generated (e.g. domain randomization [52, 38]), and also several algorithms that a practitioner can come up with in order to solve the task (e.g. style transfer). In order to narrow the scope and propose a formal solution to the problem, we propose to interpret this problem from a DA perspective. The goal of this learning paradigm is to deal with the lack of labeled data in a target domain (e.g real world) by transferring knowledge from a labeled source domain (e.g. virtual word). Therefore, clearly applicable to our problem setting.

In order to properly formalize this view, we must add a few extra assumptions and notation. Specifically, we assume that both the source inputs $x_i^s$ and target inputs $x_i^t$ are sampled i.i.d. from distributions $P_s$ and $P_t$ respectively, both over $\mathcal{X}$. We assume the output domain be $\mathcal{Y} = \{0, 1\}$ and define the indicator of performance to be the risk of a hypothesis $h : \mathcal{X} \to \mathcal{Y}$ w.r.t. the labeling function $f$, using a loss function $\ell : \mathcal{Y} \times \mathcal{Y} \to \mathbb{R}_+$ under distribution $\mathcal{D}$, or formally $R_{\mathcal{D}}^{\ell}(h, f) := \mathbb{E}_{\mathcal{D}}[\ell(h(x), f(x))]$. We let the labeling function be the optimal Bayes classifier $f(x) = \arg\max_{\hat{y} \in \mathcal{Y}} P(y = \hat{y}|x)$ [35], where $P(y|x)$ denotes the class conditional distribution for either the source-domain (simulator) $(P_s(y|x))$ or the target domain (real-world) $(P_t(y|x))$. For simplicity, we define $R_S^{\ell}(h) := R_{P_s}^{\ell}(h, f_s)$ and $R_T^{\ell}(h) := R_{P_t}^{\ell}(h, f_t)$.

For reasons that will become obvious later (Sec. 4), we will interpret the hypothesis $h$ (model) as the composition of $h = \hat{h} \circ g$ with $g : \mathcal{X} \to \mathcal{Z}$, and $\hat{h} : \mathcal{Z} \to \mathcal{Y}$, where $\mathcal{Z}$ is a representation space. Thus, we define the hypothesis class $\mathcal{H} := \{\hat{h} \circ g : \hat{h} \in \hat{\mathcal{H}}, g \in \mathcal{G}\}$ such that $h \in \mathcal{H}$. With this in hand, we can use the generalization bound from [3] to measure the generalization performance of a model $h$:

**Theorem 1** *(Acuna et al. [3]). Suppose* $\ell : \mathcal{Y} \times \mathcal{Y} \to [0, 1]$ *and denote* $\lambda^* := \min_{h \in \mathcal{H}} R_S^{\ell}(h) + R_T^{\ell}(h)$. *We have:*

$$R_T^{\ell}(h) \leq R_S^{\ell}(h) + D_{h,\mathcal{H}}^{\phi}(P_s||P_t) + \lambda^*, \quad where \tag{1}$$

*where:* $D_{h,\mathcal{H}}^{\phi}(P_s||P_t) := \sup_{h' \in \mathcal{H}} |\mathbb{E}_{x \sim P_s}[\ell(h(x), h'(x))] - \mathbb{E}_{x \sim P_t}[\phi^*(\ell(h(x), h'(x)))]|,$ *and the function* $\phi : \mathbb{R}_+ \to \mathbb{R}$ *defines a particular* $f$*-divergence with* $\phi^*$ *being its (Fenchel) conjugate.*

Theorem 1 is important for our analysis as it shows us what we have to take into account such that a model can generalize from the virtual to the real-world. Intuitively, the first term in the bound accounts for the performance of the model on simulation, and shows us, first and foremost, that we must perform well on $S$ (this is intuitive). The second term corresponds to the discrepancy between the marginal distributions $P_s(x)$ and $P_t(x)$, in simpler words, how dissimilar the virtual and real world are from an observer's view (this is also intuitive). The third term measures the ideal joint hypothesis $(\lambda^*)$ which incorporates the notion of adaptability and can be tracked back to the dissimilarity between the labeling functions [3, 59, 4]. In short, when the optimal hypothesis performs poorly in either domain, we cannot expect successful adaptation. We remark that this condition is necessary for adaptation [5, 4, 3, 59].

**Comparison vs standard DA.** In standard DA, two main lines of work dominate. These mainly depend on what assumptions are placed on $\lambda^*$. The first group (1) assumes that the last term in equation 1 is negligible thus learning is performed in an adversarial fashion by minimizing the risk in source domain and the domain discrepancy in $\mathcal{Z}$ [16, 3, 33, 57]. The second group (2) introduces reweighting schemes to account for the dissimilarity between the label marginals $P_s(y)$ and $P_t(y)$. This can be either implicit [24] or explicit [32, 50, 48]. In our scenario the synthetic dataset is sampled from the self-driving simulator. Therefore, we have control over how the data is generated, what classes appear, the frequency of appearance, what variations could be introduced and where the assets are placed. Thus, Theorem 1 additionally tell us that the data generation process must be done in such a way that $\lambda^*$ is negligible, and if so, we could focus only on learning invariant-representations. We exploit this idea in Section 3 for the data generation procedure, and in Section 4 for the training algorithm. We emphasize the importance of controlling the dissimilarity between the label marginals

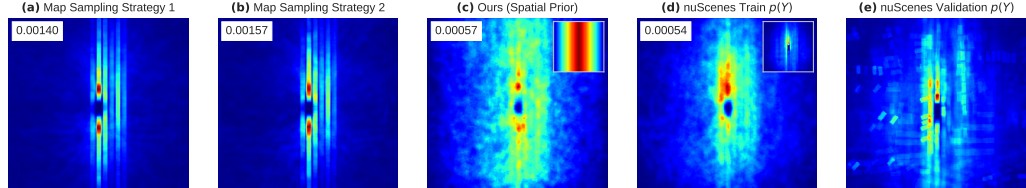

Figure 2: **Marginals induced by different sampling strategies for the task of BEV vehicle segmentation**. Figures (a) and (b) show the induced marginal $P_s(y)$ when the sampling strategy follows standard approaches: e.g. the NPC locations are sampled based on the structure of the road or the map's drivable area (pseudocode in appendix). Figure (c) shows the induced marginal when the NPC locations are sampled using our spatial prior (see Sec.3.1). Figure (d) is for reference as it assumes access to target labeled data. It shows the induced marginal when the NPC locations are sampled using a target prior estimated on the nuScenes training set (see Sec.3.2). Figure (e) shows the estimated label marginal of the nuScenes validation set for the task of vehicle BEV segmentation $P_t(y)$. The number on the top-left of the figures corresponds to the JSD between the induced marginal, and the nuScenes validation set estimated marginal (lower is better). In comparison with standard approaches, our sampling strategy minimizes the divergence between the task label marginals (e.g. $5.7e^{-4}$ vs $1.40e^{-3}$) and leads to a more diverse synthetic dataset. Notably, comparing (c) and (d), we can observe their distances are in the same order, yet sampling based on the spatial prior does not require access to labeled data.

$P_s(y)$ and $P_t(y)$, as it determines the training strategy that we could use. This is illustrated by the following lower bound from [59]:

**Theorem 2** *(Zhao et al. [59]) Suppose that* $D_{\text{JS}}(P_s(y)\|P_t(y)) \geq D_{\text{JS}}(P_s(z)\|P_t(z))$ *We have:* $R_T^\ell(h) + R_S^\ell(h) \geq \frac{1}{2}\left(\sqrt{D_{\text{JS}}(P_s(y)\|P_t(y))} - \sqrt{D_{\text{JS}}(P_s(z)\|P_t(z))}\right)^2$ *where* $D_{\text{JS}}$ *corresponds to the Jensen-Shanon divergence.*

Theorem 2 is important since it is placing a lower bound on the joint risk, and it is particularly applicable to algorithms that aim to learn domain-invariant representations. In our case, it illustrates the paramount importance of the data generation procedure in the simulator. If we deliberately sample data from the simulator and position objects in a way that creates a mismatch between real and virtual world, simply minimizing the risk in the source domain and the discrepancy between source and target domain in the representation space may not help. Simply put, failing to align the marginals may prevent us from using recent SoTA adversarial learning algorithms.

## 3 Synthetic Data Generation

In the previous section, we analyzed learning from a simulator from a domain adaptation perspective, and showed the importance of the data sampling strategy as this could create a mismatch between the label marginals. In this section, we take insipiration from this analysis and propose two simple, but effective methods to sample data from the self-driving simulator. The first one proposes the use of spatial priors and is targeted to a regime where there is a complete absence of real-world labeled data. The second one assumes that a few datapoints are labeled in the target domain and exploits these labels to estimate a prior for the positions of the vehicles in the map.

### 3.1 Sampling with Spatial Priors

If labels in the target domain are not available, we cannot measure the divergence between the labels marginals $P(y)$ between the generated and the real world data. That said, given the bird's-eye view segmentation map, it is not hard for a human to design spatial priors representing locations with high probability of finding a vehicle. As visualized in Figure 2, we design a simple prior that samples locations for non-player characters (NPCs) proportional to the longitudinal distance of the vehicle to the ego car. For position $(x_1, x_2) \in \mathbb{R}^2$ in meters relative to the ego car:

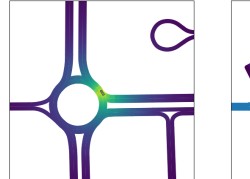 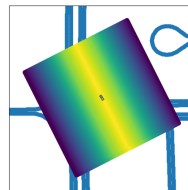

Figure 3: **Sampling Methods (Left)** Standard approaches sample NPC locations from the map's drivable area based on the structure of the road. **(Right)** We sample the NPC locations based on our spatial prior (independent of the structure of the road) which aims for diversity in the generated data and minimizes the divergence between the task label marginals (see also Fig 2 ).

$$P_{\text{spatial}}(x_1, x_2) \propto \begin{cases} -\frac{1}{125}|x_2| + 0.6 & |x_2| \leq 12.5 \\ -\frac{1}{75}(|x_2| - 50) & |x_2| > 12.5 \end{cases} \quad (2)$$

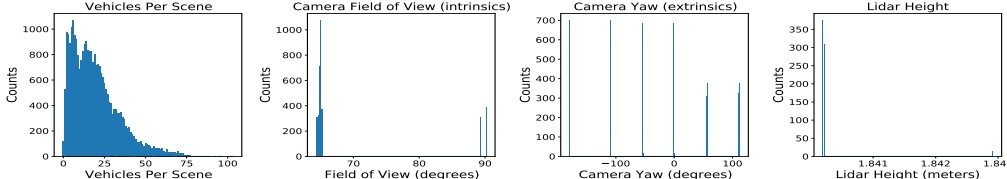

Figure 4: **Matching nuScenes Statistics** From left to right, we show the distribution of number of vehicles per scene in nuScenes, the distribution of field of view of the nuScenes cameras, the yaw relative to the ego coordinate frame of the cameras (6 peaks for the 6 different camera directions), and the height of the LiDAR (roughly the same across all scenes). These statistics are matched when sampling data from CARLA.

These numbers correspond to linearly interpolating between a probability proportional to $0.0$ for $|x_2| = 50$ meters, $0.5$ for $|x_2| = 12.5$ meters, and $0.6$ for $|x_2| = 0$ meters. Intuitively, the prior models the fact that other vehicles are generally located along the length-wise axis of the ego vehicle. The density is independent of $x_1$ which models our prior that on a straight road, a vehicle is equally likely to be $\hat{x}_1$ meters in front or behind the ego, for any $\hat{x}_1 \in \mathbb{R}$.

## 3.2 Sampling based on Target Priors

Often in practical applications, a small proportion of data is available in the target domain. In such scenarios a natural prior for sampling would be to compute $P_t(y)$ (see Figure 2 for an example), and generate the data based on that. Depending on how large the amount of data in the target domain is, we could also create a blend $\alpha P_{\text{spatial}} + (1 - \alpha)P_t$ and sample NPC locations from it, for any $\alpha \in [0, 1]$. We emphasize that we do not use sampling based on target priors in our experiments since our work focuses on the challenging setting where labeled data is not available in target domain.

## 4 Training Strategy

In the previous section, we designed a simulator-agnostic sampling strategy that builds on the intuition of minimizing the distance between the label marginals. From Theorem 1, we can also observe that in order to improve performance in the real world, we must minimize the distance between the input distributions $P_s(x)$ and $P_t(x)$. In this section, we extend the training algorithm from [3] to dense prediction tasks, with the aim to learn domain-invariant representations and minimize the divergence between virtual and real world in a latent space $\mathcal{Z}$. Minimizing the divergence in a representation space allows domain adaptation algorithms to be sensor and architecture agnostic. We additionally take inspiration from [47] and incorporate the use of pseudolabels into the $f$-DAL framework.

We remark in our scenario we cannot effectively learn invariant representations using adversarial learning unless the data sampling strategy from Section 3 is used because otherwise there may be a misalignment between the task label marginals (see Section 2 and [3, 59] for more details).

**Learning Domain-Invariant Representations**. Our training algorithm can be interpreted as simultaneously minimizing the source domain error and aligning the two distributions (virtual and real worlds) in some representation space $\mathcal{Z}$. Formally, we aim at finding a hypothesis $h$ that jointly minimizes the first two terms of Theorem 1. Let $h : \mathcal{X} \rightarrow \mathcal{Y}$ with $\mathcal{Y} \subseteq \mathbb{R}^{h \cdot w}$ which corresponds to a pixel-wise binary segmentation map of dimensions $h$ and $w$ respectively, with $\mathcal{X}$ corresponding to an unordered set of multi-view images for camera-based bird's-eye-view segmentation $\mathcal{X} \subseteq \mathbb{R}^{N \cdot 3 \cdot H \cdot W}$, and a point cloud for the lidar-based bird's-eye-view segmentation $\mathcal{X} \subseteq \mathbb{R}^{3 \cdot N}$. Our objective function is then formulated in an adversarial fashion by minimizing the following objective:

$$\min_{\hat{h} \in \hat{\mathcal{H}}, g \in \mathcal{G}} \max_{\hat{h}' \in \hat{\mathcal{H}}} \mathbb{E}_{x, y \sim p_s}[\ell(\hat{h} \circ g, y)] + d_{st}, \tag{3}$$

where $\ell : \mathbb{R}^{h \cdot w} \times \mathbb{R}^{h \cdot w} \rightarrow \mathbb{R}_+$ is defined as: $\ell(p, q) := \frac{1}{h \cdot w} \sum_{i=0}^{h \cdot w} \beta p_i \log q_i + (1 - p_i) \log(1 - q_i)$, and $p_i$ and $q_i \in [0, 1]$ (the averaged binary-cross entropy loss). $\beta$ is a hyperparameter that weights the importance of positive pixels in the pixel-wise binary segmentation map. We define $d_{st}$ as:

$$d_{st} := \mathbb{E}_{x \sim p_s} \left[ \mathbb{E}_{h \cdot w}[(\hat{h}' \circ g)] \right] - \mathbb{E}_{x \sim p_t} \left[ \mathbb{E}_{h \cdot w}[\frac{1}{4}(\hat{h}' \circ g)_i^2 + \hat{h}' \circ g(x)_i] \right] \tag{4}$$

which corresponds to the Pearson $\chi^2$ divergence, and $\mathbb{E}_{h \cdot w}[x] := \frac{1}{h \cdot w} \sum_{i=0}^{h \cdot w} x_i$. Different from the original formulation of $f$-DAL Pearson [3], $d_{st}$ can be interpreted in our case as a per-location-domain-classifier. For simplicity, we use the gradient reversal layer from [16, 15] to deal with the min-max objective in a single forward-backward pass.

**Pseudo-Labels.** In addition to the objective in Equation 3, we use the following pseudo-loss:

$$\ell_{\text{pseudo}}(p, p_{\text{aug}}) := \sum_{i=0}^{h \cdot w} \mathbb{1}[p_i \geq \tau][\beta \log p_{\text{aug}_i}] + \mathbb{1}[1 - p_i \leq 1 - \tau] \log(1 - p_{\text{aug}_i}) \qquad (5)$$

where $\tau \in [0, 1]$, $\mathbb{1}[x]$ corresponds to the indicator function, $p := h \circ g(x^t)$ and $p_{\text{aug}} := h \circ g \circ \text{aug}(x^t)$ with aug $: \mathcal{X} \to \mathcal{X}$ a function that produces a strong augmentation on the input data point. For camera sensors, we let aug be a version of RandAugment [10] that additionally incorporates camera dropping. For lidar, we follow the same idea from [10] but replace the image augmentations by random noise over the points positions as well as points dropout. We let $\tau = 0.9$ in all our experiments. More details are provided in the supplementary material.

The objective in Equation 5 is inspired by [47]. Intuitively, it is based on the generation of pseudo-labels using the confident pixels in the model's predictions, as determined by $\tau$, on the target domain, on a non-strongly augmented version of the same data point. Therefore, it explicitly exposes the model during training to real-world data. We experimentally observed that using pseudo-labels without enforcing invariant representations performs worse than a simple vanilla model trained only on the source dataset, likely because as opposed to [47] in our scenario the target/unlabeled data comes from different distributions. The use of pseudo-labels and adversarial learning in our algorithm can be justified through the results of [45] and Theorem 1. In summary, our training algorithm jointly minimizes equations 3 and 5.

**Other theoretically inspired training strategies.** In principle several other training algorithms that minimize the discrepancy between source and target can be used. For example, style transfer using MUNIT [23], the original formulation of [16] or $f$-DAL [3]. Experimentally, we found our generalization of $f$-DAL-Pearson and the use of pseudo-labels to be more performant (see Table 3). We hypothesise the reason for domain-invariant representation methods being better is because these minimize the dissimilarity between source and target in a low dimensional space. In contrast, style transfer approaches operate directly on higher dimensional input data (e.g. images). Moreover, the use of pseudo-labels further exposes the model to training examples on real-world data.

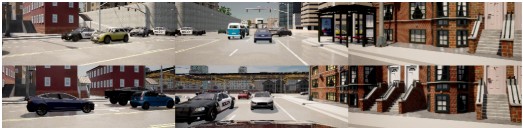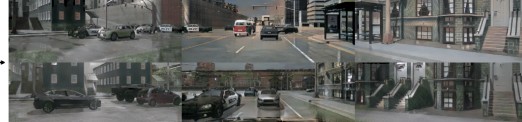

Figure 5: **Style Transfer Baseline (MUNIT)** For our style transfer baseline, we sample a style indepedently for each image and "translate" the CARLA images into nuScenes as shown above. The Lift Splat model receives images like those on the right as input during training.

## 5 Related Work

**Synthetic Data Generation.** The use of synthetic data as an alternative to dataset collection and annotation has received significant interest in recent years. Synthetic data with labels can be generated in different ways, e.g., with generative models such as GANs [58, 31], data-driven reconstruction with sensor simulation [34, 8, 27], or via the use of graphics simulators. We here focus on the latter.

Various graphics-rendered synthetic datasets have been created for tasks such as object detection, semantic segmentation, and home robotics among many others [39, 36, 14, 44, 41]. Several techniques have been proposed to generate useful labeled data from the graphics engine, informed by the target real data. For example, [52] proposed to randomize the parameters of the simulator in non-realistic ways with the aim of forcing the neural network to learn the essential features of the object of interest. In a similar vein, [38] proposes to procedurally generate synthetic images while preserving the structure, or context, of the problem at hand. Instead of randomizing, [25, 11, 30] proposed a data-driven approach. Specifically, the authors aim to resemble a target dataset by searching over set of parameters of a surrogate function that interfaces with a synthesizer. These methods are all

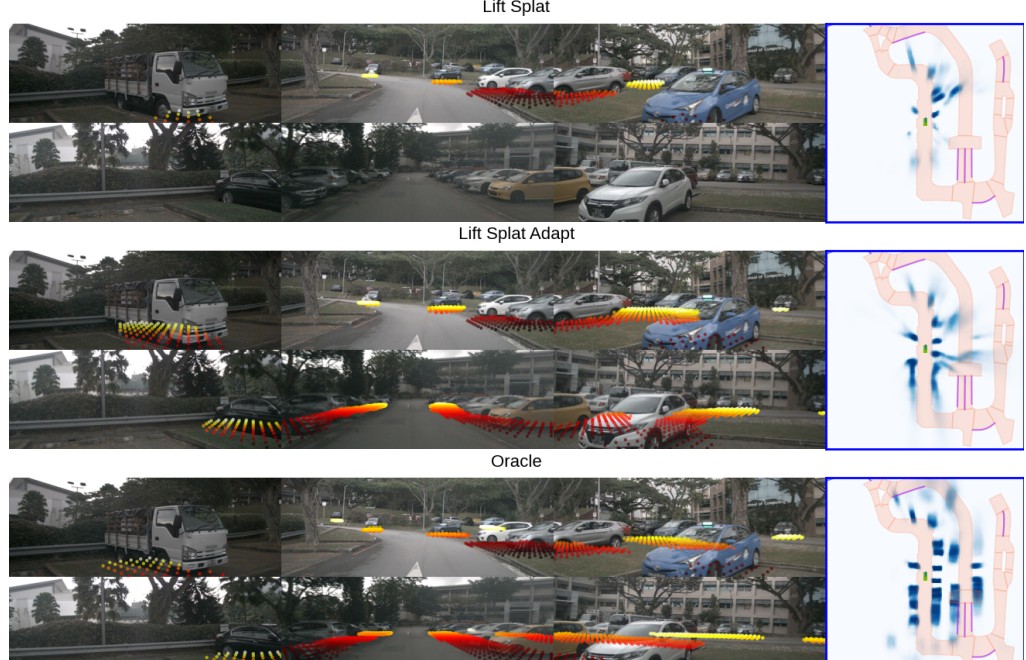

Figure 6: **Lift Splat Adapt** We visualize the output of Lift Splat (LS) for a single timestep from nuScenes val. The model takes as input the 6 images shown on the left, outputs the heatmap shown on the right, and we project the prediction back onto the input images colorized according to depth. Top is a LS model without adaptation, middle is a LS model trained with our method, and bottom is a LS model trained directly on nuScenes.

different from us because 1) they require access to the procedural model of the simulator, thus they are not easy to apply to off-the-self self-driving simulators, 2) they are focused on the generation of single dashcam RGB images instead of a sensor suite, and 3) data-driven approaches could require a significant amount of data in the target domain. Most importantly, the transfer performance that a practitioner may observe lacks justification and guidance, e.g. methods such as domain randomization may increase the divergence between the label marginals if not carefully done.

**Domain Adaptation on Synthetic Data.** There is a significant body of work on domain adaptation ranging from theory and algorithms to applications [4, 57, 3, 16, 15, 21], some of them applicable to sim-to-real datasets such as [44, 41] and tasks such as semantic segmentation and object detection [18, 26, 53]. Most of the literature in this direction however analyzes the problem with a fixed dataset perspective. They assume that the source and target datasets are given and aim to find way to minimize the gap. In our formulation, we have access to the self-driving simulator. Therefore, we have control over how the data is generated, what variations are introduced and where the objects are placed. Our method thus is a unified view that minimizes the discrepancy between source and target distribution by building on previous work such as [17, 3] and ensuring the effectiveness of them through a sampling strategy that aims at minimizing the distance between the labels' marginals.

## 6 Experimental Section

In this section we quantify the performance of the proposed data-generation and training strategies using an open source self-driving simulator (Carla, MIT license [13]) and a real-world dataset (nuScenes, Apache license [7]) . We first introduce the self-driving simulator and the data generation setup. We then discuss the methods in comparison and our proposed baselines. In Section 6.2, we show our main experimental results on the task of bird-eye-view (BEV) vehicle segmentation. Finally, we provide an extensive experimental analysis where we ablate our data and training strategy, and aim to understand what defficiencies of the self-driving simulator can be compensated with our approach.

### 6.1 Self-Driving Simulator and Methods in Comparison

**Self-Driving Simulator.** We use CARLA version 0.9.11 as the self-driving simulator. For the synthetic data generation, we sample nuScenes-like datasets from CARLA such that dataset consists of 4000 episodes; each episode lasts 4 seconds during which all vehicles are controlled by CARLA's

default traffic manager. 3D bounding boxes and images are stored synchronously at 2hz [54], and LiDAR scans are stored at 20hz as in nuScenes [7]. We also store the ego pose to facilitate accumulating LiDAR scans across multiple timesteps. Each CARLA dataset is 260 GB. We split each dataset into a training set of 28k timesteps (which matches the number of timesteps in the nuScenes training set) and a validation set of 4k timesteps. The datasets and scripts for generating them will be publicly released. In the base case for all datasets, we match the distribution of number of vehicles per episode and the distribution of camera and lidar extrinsics and intrinsics to nuScenes (see Fig 4).

**Methods in Comparison.** For the Synthetic Data Generation strategy, we propose a baseline where we sample data based on the road structure, and randomize what is possible in the simulator, e.g. sampling the color of vehicle assets or weather parameters independently and uniformly. This method is inspired by [52, 38] but within the realistic restriction of the self-driving simulator. We refer to this strategy as **RS** (road structure). We also add extra baselines on top of RS to account for the domain-

Table 1: Lift-Splat Sim → Real. We compare the performance of our method vs RS. To account for the reality gap, we also show results of RS with different adaptation techniques.

| Method | IOU |
|---|---|
| RS-No Adaptation | 9.76 |
| RS-Ensemble (Inspired by [12]) | 11.95 |
| RS-Style Transfer (MUNIT) | 13.33 |
| RS-DANN | 13.76 |
| RS-Ensemble + Test-time Aug (Inspired by [12]) | 13.76 |
| Ours | **17.84** |

gap: e.g. style transfer with MUNIT [23] and using a domain adaptation method inspired by [16]. Details on the MUNIT architecture can be found in the supplementary. We refer to these as **RS-Style-Transfer** and **RS-DANN** respectively. If adaptation is not used we refer to it **RS-No-Adaptation**. Finally, we have two additional baselines inspired by [12] which we call **RS-Ensemble** and **RS-Ensemble + Test-time Aug**. Since these ensemble baselines use ground-truth to choose the prediction, they represent an upper-bound on the performance of ensemble baselines used in the Waymo adaptation challenge [12]. More details in the supplementary material.

## 6.2 Bird's-Eye-View Vehicle Segmentation

Our main experiments and analysis are based on the task of BEV vehicle segmentation from multi-camera sensor data. We additionally compare to baselines using Lidar observations.

**Camera.** In this experiment, the input data corresponds to an unordered set of multi-view images. We use Lift Splat method [37]. We use the same backbone [49] and training scheme [28, 9] as in [37].

Table 2: Point Pillars Sim → Real. We compare performance of our method vs RS.

| Method | IOU |
|---|---|
| RS-No Adaptation | 15.09 |
| RS-DANN | 14.41 |
| Ours | **17.20** |

**Lidar.** In this experiment the input data corresponds to point cloud obatained from the lidar scan. We additionally show our method can be used in a completely different sensor such as Lidar Data. For the network architecture, we follow Point Pillars [29], a standard LiDAR architecture consisting of a shallow pointnet [40] followed by a deep 2D CNN based on resnet18 [20].

**Semi-Supervised Scenario.** We also show experiments if labeled real world data is available. In Figure 8, we plot transfer performance when we use 2%, 5%, and 25% of nuScenes training set.

## 6.3 Analysis

In this section, we first ablate the choice of the training strategy and then analyze the correlation between transfer performance and the distance between the labels' marginals as this constitutes the main motivation behind our sampling strategy. We then analyze functionality of the self-driving simulator to improve transfer performance and what deficiencies can be compensated with adaptation.

**Training Strategy** In Table 3 we keep the data generation strategy fixed and evaluate the performance of different training strategy such as Style-Transfer, DANN, our generalized version of $f$-DAL, and $f$-DAL-Pearson with pseudo-labels as proposed in section 4. Our method achieves the best results. Similar to what was found [3], we observe $f$-DAL performs better than DANN. We also observe that domain-adversarial approaches perform better than Style-Transfer. We believe this is because it is easier to close the reality gap in a latent space rather than in the very high dimensional image space.

Table 3: Ablation on the Training Strategy. In this scenario, we fix the data-generation strategy to be the best.

| Name | IOU |
|---|---|
| No Adaptation | 10.33 |
| Style Transfer (MUNIT) | 14.32 |
| DANN | 14.45 |
| $f$-DAL Jensen | 16.19 |
| $f$-DAL Pearson | 17.03 |
| Ours | **17.84** |

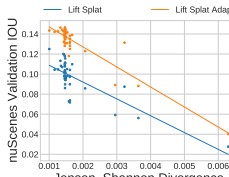

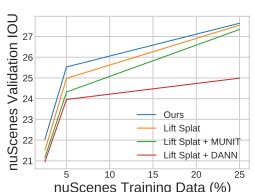

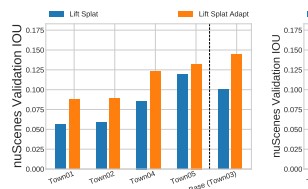

Figure 7: **IOU vs. JSD** Transfer performance is negatively correlated with the distance between $P_s(y)$ and $P_t(y)$ as expected from Theorem 2.

Figure 8: **Semi-Super. Learning** Our method improves over baselines when $p\%$ of the labeled data is available at training.

Figure 9: **Better Sampling Improves Town Robustness** We compare transfer performance of Lift Splat models when NPCs are sampled according to geometry of the town roads (left) vs the hard-coded prior (right). Performance greatly improves when using a sampling strategy that doesn't condition on road structure.

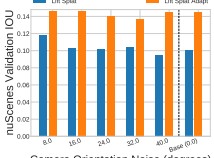

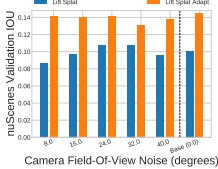

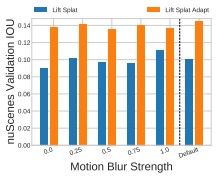

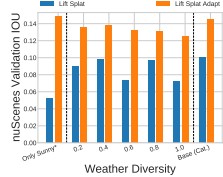

Figure 10: **Camera Intrinsics and Extrinsics** Instead of sampling camera extrinsics and intrinsics that exactly correspond to extrinsics and intrinsics from nuScenes, we sample yaw (left) and field of view (middle) from nuScenes with some variance.

Figure 11: **Weather** variation does not appear to be a limiting factor in our setting.

**Transfer performance vs JSD** We also evaluate the distance between the label marginals computed as $D_{\mathrm{JS}}(P_s(y)||P_t(y))$ vs transfer performance (IoU). Datasets with a smaller distance perform better.

**Maps** When simulating with CARLA, different "towns" can be chosen. Each town has a different road topology and different static obstacles like buildings and foliage. In Figure 9, we show that performance is sensitive to the town if a simple sampling strategy based on the road structure is used. However, by sampling NPC locations to minimize the JSD in the marginals, we increase performance by a large amount.

**Vehicle Assets** An intuitive goal for self-driving simulators has been to improve the quantity and quality of vehicle assets [55, 19]. In Figure 12, we show that performance is sensitive to the number of vehicle assets used in the simulator and the number of NPCs sampled per episode, but is not so sensitive to the variance in the colors chosen for the NPCs.

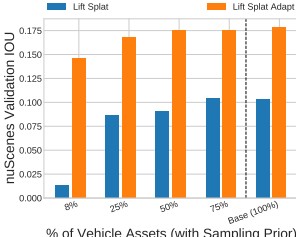

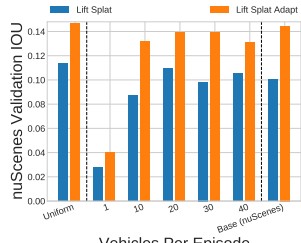

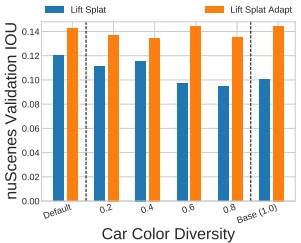

Figure 12: **Vehicle Assets** (Left) We vary the number of vehicle assets used when sampling CARLA data. Interestingly, Lift-Splat-Adapt (ours) is able to compensate for performance when very few assets are used. (Middle) We compare performance when the number of NPCs per episode is sampled from $\mathrm{Unif}(0, 40)$, fixed at 1/10/20/30/40 NPCs per episode, or sampled from the distribution in Figure 4. Finally, we compare performance on datasets in which car colors are independently sampled from $\mathrm{Unif}(0, x)$ vs. default car colors.

**Weather** nuScenes contains sunny, rainy, and nighttime scenes. We test the extent to which it's important for CARLA data to also have variance in weather in Figure 11. CARLA has 15 different "standard" weather settings as well as controls for cloudiness, precipitation, precipitation deposits (e.g. puddles), wind intensity, wetness, fog density, and sun altitude angle (which controls nighttime vs. daytime on a sliding scale). We randomly sample these controls from $x \sim \mathrm{Unif}(0, v)$ for $v \in \{0.2, 0.4, 0.6, 0.8, 1.0\}$ and compare performance against sampling categorically from the 15 preset weather settings. We also compare against exclusively using the "sunny" weather setting but sampling the location of the NPCs according to the hard-coded prior. We find that our method is able to compensate for much of the loss in weather diversity and using only the sunny weather setting on

data where NPCs are sampled according to the prior achieves higher performance than any amount of weather diversity with NPCs sampled according to the town.

**Camera Post-Processing** The CARLA "scene final" camera provides a view of the scene after applying some post-processing effects to create a more realistic feel. Specifically, CARLA applies vignette, grain jitter, bloom, auto exposure, lens flares and depth of field as post-process effects [1]. In figure 7, we evaluate the importance of this post-processing stage. We can see adaptation is able to compensate for the loss in realism by a large amount.

## 7   Limitations and Societal Impact

**Limitations** In Figure 15, we qualitatively demonstrate certain limitations of our method. It is unlikely that adaptation can be used to fix errors for vehicles that look significantly different from the vehicle assets used in CARLA. We also found that our adapted model misdetects pedestrians for vehicles occasionally, perhaps because it has learned a non-causal correlation between legs and bicycles. Finally, the adapted model occasionally midpredicts the depth of vehicles.

**Societal Impact** In this paper, we primarily seek to optimize the intersection-over-union of bird's-eye-view segmentation models by leveraging synthetic data rendered in CARLA. While simulated data can be controlled to mitigate some of the biases of real world data (for instance by uniformly sampling vehicles of any color), synthetic data also brings it's own biases in that not all materials, lighting conditions, or weather patterns can be simulated equally realistically; snow is notoriously difficult to simulate correctly, for instance, so simulated data may not be the solution for improving perception in snowy climates. We believe using simulation will be crucial for training and testing safe self-driving systems that have the potential to make public roads safer and more efficient.

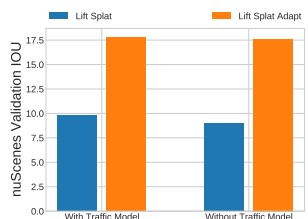

Figure 13: **Traffic Model** Performance is unaffected by whether or not a traffic model is used when generating synthetic data.

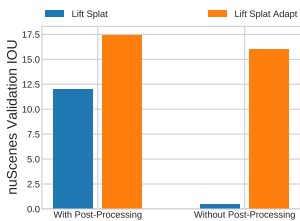

Figure 14: **Camera Post-Processing** CARLA uses post-processing to improve the realism of images. We evaluate the importance of this post-processing by toggling "enable postprocess effects". Adaptation is able to compensate for the loss in realism by a large amount.

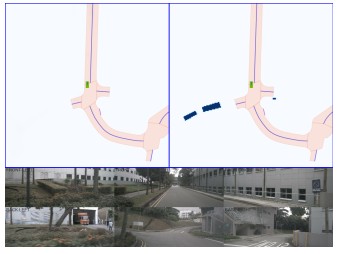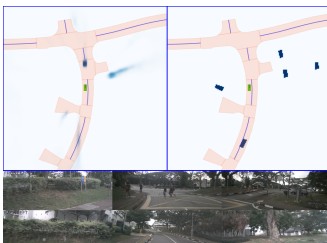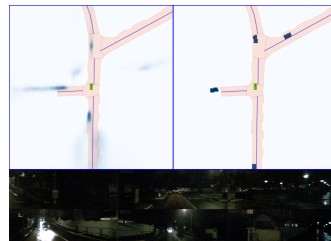

Figure 15: **Limitations** Predictions of the best Lift Splat Adapt model vs. ground-truth when the model makes an error. (Left) The model does not detect the orange bus in the garage. (Middle) The model predicts a vehicle where there are pedestrians. (Right) The model predicts a car behind it is much closer than it actually is.

## 8   Conclusion

Using recent advances in domain adaptation, we motivate methods for both sampling and for training on synthetic data such that models transfer well to real world data. We demonstrate the effectiveness of our method by training camera-based and lidar-based bird's-eye-view vehicle segmentation models on data sampled from the CARLA simulator and evaluated on real world data from nuScenes.

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
