# 9 Supplementary Material

## 9.1 Lift-Splat Adapt Diagram

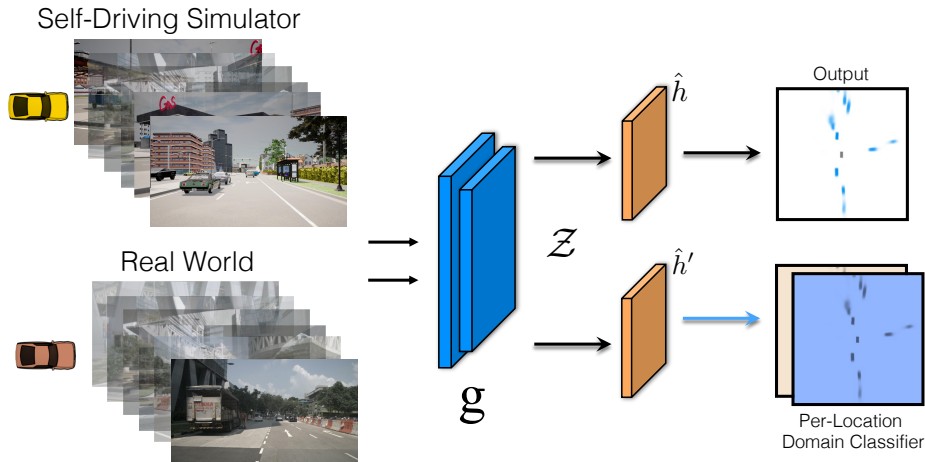

Figure 16: Lift-Splat-Adapt Architecture. Lift-Splat-Adapt follows the same architecture from Lift-Splat-Shoot [37]. To compute the discrepancy term $d_{st}$, we add a per-location domain classifier $\hat{h}'$. This is introduced before the final up-sampling module of Lift-Splat's BevEncoder. It constitutes two Conv layers with LeakyRelu non-linearity that predicts whether a pixel in the $H \times W$ semantic map corresponds to the either source or target domain. On the other hand, $\hat{h}$ predicts the Bird-Eye View binary segmentation map. $g$ constitutes Lift-Splat backbone, i.e. CamEncoder and BevEncoder (up to the last up-sampling module).

In figure 9.1 we show the Lift-Splat Adapt diagram. Our training strategy requires little modification to the original architecture, e.g. only the per-location domain classifier $\hat{h}'$ is added on top. $\hat{h}'$ constitutes two Conv layers with LeakyRelu non-linearity that predicts whether a pixel in the $H \times W$ semantic map corresponds to either source or target domain. We also experimented with $\hat{h}'$ being similar to $\hat{h}$ following the recommendation from [3] and observed very similar performance (slightly better in our case).

## 9.2 Training and Architecture Hyperparameters

**MUNIT** We use the official MUNIT pytorch codebase `https://github.com/NVlabs/MUNIT`. We use the config parameters below.

When using MUNIT to improve domain adaptation, we sample style vectors independently for each camera on-the-fly. We include a video of example translations.

```
# logger options
image_save_iter: 10000
image_display_iter: 100
display_size: 16              # How many images do you want to display each time
snapshot_save_iter: 10000     # How often do you want to save trained models
log_iter: 1                   # How often do you want to log the training stats

# optimization options
max_iter: 1000000             # maximum number of training iterations
batch_size: 1                 # batch size
weight_decay: 0.0001          # weight decay
beta1: 0.5                    # Adam parameter
```

```
beta2: 0.999                      # Adam parameter
init: kaiming                     # initialization [gaussian/kaiming/xavier/orthogonal]
lr: 0.0001                        # initial learning rate
lr_policy: step                   # learning rate scheduler
step_size: 100000                 # how often to decay learning rate
gamma: 0.5                        # how much to decay learning rate
gan_w: 1                          # weight of adversarial loss
recon_x_w: 10                     # weight of image reconstruction loss
recon_s_w: 1                      # weight of style reconstruction loss
recon_c_w: 1                      # weight of content reconstruction loss
recon_x_cyc_w: 10                 # weight of explicit style augmented cycle consistency loss
vgg_w: 0                          # weight of domain-invariant perceptual loss

# model options
gen:
    dim: 64                       # number of filters in the bottommost layer
    mlp_dim: 256                  # number of filters in MLP
    style_dim: 8                  # length of style code
    activ: relu                   # activation function [relu/lrelu/prelu/selu/tanh]
    n_downsample: 2               # number of downsampling layers in content encoder
    n_res: 4                      # number of residual blocks in content encoder/decoder
    pad_type: reflect             # padding type [zero/reflect]
dis:
    dim: 64                       # number of filters in the bottommost layer
    norm: none                    # normalization layer [none/bn/in/ln]
    activ: lrelu                  # activation function [relu/lrelu/prelu/selu/tanh]
    n_layer: 4                    # number of layers in D
    gan_type: lsgan               # GAN loss [lsgan/nsgan]
    num_scales: 3                 # number of scales
    pad_type: reflect             # padding type [zero/reflect]

# data options
input_dim_a: 3                                # number of image channels [1/3]
input_dim_b: 3                                # number of image channels [1/3]
num_workers: 7                                # number of data loading threads
crop_image_height: 128                        # random crop image of this height
crop_image_width: 352                         # random crop image of this width
```

**Lift Splat** We use the same training parameters from the official release of "Lift, Splat, Shoot" https://github.com/nv-tlabs/lift-splat-shoot. We use the default train/validation split from nuScenes. All numbers reported in the paper are on the validation split of nuScenes. Images are randomly resized by an amount uniformly chosen between $(0.193, 0.225)$, then cropped and padded to make the dimensions $128 \times 352$ before being fed to the network. During training, we clip the L2 norm of the gradients by $5.0$ and weighr positive examples in the heat map by $2.13$. The bird's-eye-view grid is $x = (-50.0, 50.0, 0.5)$ and $y = (-50.0, 50.0, 0.5)$. Depth during the lift step is discretized by $d = (4.0, 45.0, 1.0)$. We use the Adam optimizer with learning rate 1e-3 and weight decay 1e-7 and train for 50 epochs with a batch size of 4 (350k steps) using an internal cluster of V100 GPUs. We include a video of example predictions for "Lift Splat" and our best "Lift Splat Adapt" model.

**Adapt Version (Images)** To solve the minimax objective in a single forward-backward pass we use the gradient-reversal layer GRL and warm-up schedule from [16, 15]. Specifically, we set the warm-up to reach its maximum value $0.78$ after 570 iterations. We add a coefficient $\lambda$ to up-weight the importance of $d_{st}$ in the loss in equation 3 and set it to $1.87$. We mix the same amount of source and target samples (e.g. 5 images) in a batch (4 samples source and 4 samples target). The pseudo-loss coefficient $\tau$ is set to 0.9. We train for 35 epochs and use a learning rate of 0.01 with polynomial decay (0.70). We use SGD with Nesterov Momentum (0.9). We observed Adam was be unstable in this scenario. We found proper tuning of the warm-up coefficients for the GRL and Lift-Splat $\beta$ parameter (equation 3) to be important. For all models (including baselines), we tune them using a grid search. All the other hyperparameters are kept the same from the no adaptation version.

**Point Pillars** We use the spconv library to voxelize 2 lidar scans that have been transformed into the same coordinate frame using the vehicle pose at each of the timesteps the scan was taken. In the first

stage, we extract the coordinates of each point in the point cloud relative to the centroid of the points in the pillar, relative to the center of the pillar, the reflectance of the point (normalized between 0 and 1) and the difference between the time of the LiDAR scan and the current time. In the second stage, we apply a single layer ReLU pointnet that produces a 64-dimensional latent vector. We then feed the $B \times 64 \times X \times Y$ tensor through the same bird's-eye-view decoder from the open-source Lift Splat implementation. We train point pillars with the same hyperparameters used to train Lift Splat: Adam optimizer with learning rate 1e-3 and weight decay 1e-7 and train for 50 epochs with a batch size of 4 (350k steps). We include a video of example predictions for "Point Pillars" and our best "Point Pillars Adapt" model.

**Adapt Version (Lidar)** Similar to the image version, we solve the minimax objective in a single forward-backward pass using the gradient-reversal layer GRL and the warm-up schedule from [16, 15]. In this case, we set the warm-up to reach its maximum value 0.78 after 16000 iterations. We add a coefficient $\lambda$ to up-weight the importance of $d_{st}$ in the loss in equation 3 and set it to 2.1. We mix the same amount of source and target samples in a batch (4 each). The pseudo-loss coefficient is set to 0.9. We train for 35 epochs and use a learning rate of 0.01 with polynomial decay (0.70) . We use SGD with Nesterov Momentum (0.9). We observed Adam was unstable in this scenario. All the other hyperparameters are kept the same from the original implementation of Lift-Splat.

**Strong Augmentation for Pseudo-Labels**. For camera sensors, we let aug : $\mathcal{X} \to \mathcal{X}$ be a version of RandAugment [10] with $n = 2$ (operations sampled from the pool) and $m = 10$ (values). We remove the rotation operation from the augmentation pool. In addition, we also perform camera dropping by randomly choosing 5 out of the 6 cameras with uniform probability. For Lidar, we follow the same idea of RandAugment but replace the augmentation pool for Gaussian Noise and Points Dropout. Specifically, we add Gaussian noise to the $(x, y, z)$ points position. The std of the Gaussian is uniformly chosen from the list $[0.0, 0.1, 0.2, 0.12, 0.15, 0.2, 0.25]$. We then randomly(uniform distribution) choose the percent of points to be drop from the following list $[0.0, 0.1, 0.15, 0.2, 0.3, 0.4]$ with zero meaning no dropping at all.

**More details for methods used in the comparison.** For the Synthetic Data Generation strategy, we propose a baseline where we sample data based on the road structure, and randomize what is possible in the simulator, e.g. sampling the color of vehicle assets or weather parameters independently and uniformly. This method is inspired by [52, 38] but within the realistic restriction of the self-driving simulator. We refer to this strategy as **RS** (road structure). We also add extra baselines on top of RS to account for the domain-gap. These are vs style transfer with MUNIT [23] and using a domain adaptation inspired by [16]. Details on the MUNIT architecture with example translations can be found in the supplementary. We refer to this as **RS-Style-Transfer** and **RS-DANN** respectively. We also add a version without adaptation which we refer to as **RS-No-Adaptation**. Finally, we have two additional baselines inspired by [12] which we call **RS-Ensemble** and **RS-Ensemble + Test-time Aug**. Since these ensemble baselines use ground-truth to choose the prediction, they represent an upper-bound on the performance of ensemble baselines used in the Waymo adaptation challenge[12]. Specifically, we train 4 models on the CARLA-generated data. These models individually achieve nuScenes transfer IOUs of $\{9.75, 9.88, 8.73, 9.04\}$. For the RS-Ensemble baseline, we make a prediction with each model and take the prediction with the highest IOU with respect to ground-truth. For the RS-Ensemble + Test-time Aug baseline, we randomly transform the input images according to the same augmentation parameters used in "Lift Splat"[37] and take the prediction with the highest IOU with respect to ground-truth. Note that due to the fact that these ensemble baselines use ground-truth to choose the prediction, they represent an upper-bound on the performance of ensemble baselines used in the Waymo adaptation challenge[12]

### 9.3 NPC Sampling Strategies

One can choose to sample NPC locations either using the map API that CARLA provides or by sampling locations in order to mimic a given marginal distribution. When sampling according to the map, we discretize the drivable area of the CARLA map by 1.0 meters. We provide a sample of our code for sampling $N$ NPCs according to these map locations below.

```
def map_sampling(pos_inits, nnpc, pos_agents, traffic_port):
    dists = [tr_dist(init_loc, init) for init in pos_inits]
    valid_ixes = [ix for ix in range(len(pos_inits)) if dists[ix] > 5.0]
```

```
        # weight inits closer to the ego exponentially more
        weight = np.array([np.exp(-dists[ix] / 25.0) for ix in valid_ixes])

        weight = weight / weight.sum()
        ordering = np.random.choice(valid_ixes, size=len(valid_ixes),
                                    p=weight, replace=False)

        agents = []
        for ix in ordering:
            if len(agents) == nnpc:
                break
            agentbp = np.random.choice(pos_agents)
            location = pos_inits[ix]
            agent = world.try_spawn_actor(agentbp, location)
            if agent is None:
                continue
            agent.set_autopilot(True, traffic_port)
            agents.append(agent)
        return agents
```

In order to sample NPC locations according to a given target distribution, we determine the $x, y$ locations of the pixels in the output grid of "Lift Splat" in map coordinates, then sample locations with probability determined by the given heatmap. Sample code is below.

```
def heatmap_sampling(heatmap, nnpc, pos_agents, traffic_port):
    XY = get_grid_pts()
    # need to flip here to get the coordinates to align
    XY[1] = -1 * XY[1]
    weight = heatmap.flatten() / heatmap.sum()

    ego_mat = ClientSideBoundingBoxes.get_matrix(init_loc)
    XYlocal = np.dot(ego_mat, np.concatenate((XY, np.ones((2, XY.shape[1]))), axis=0))

    count = (heatmap > 0).sum()
    ordering = np.random.choice(XYlocal.shape[1], size=count,
                                p=heatmap.flatten() / heatmap.sum(), replace=False)
    agents = []
    for ix in ordering:
        if len(agents) == nnpc:
            break
        agentbp = np.random.choice(pos_agents)
        head = np.random.uniform(0, 360.0)
        location = carla.Transform(carla.Location(x=XYlocal[0, ix], y=XYlocal[1, ix],
                                   z=XYlocal[2, ix]),
                                   carla.Rotation(yaw=head, pitch=0.0, roll=0.0))
        agent = world.try_spawn_actor(agentbp, location)
        if agent is None:
            continue
        agent.set_autopilot(True, traffic_port)
        agents.append(agent)
```