# OpenReview forum: "Towards Optimal Strategies for Training Self-Driving Perception Models in Simulation"
_NeurIPS.cc/2021/Conference — NeurIPS 2021 Poster_

### Official Review · Reviewer_gzio · 2021-07-16

**Rating:** 7
**Confidence:** 3

**Summary:**

This work targets the sim2real domain adaptation problem and approaches the problem from two angles - data collection and transfer method. The authors discuss the importance of how data is generated from the source domain and propose ways to effectively design scenarios (placement of other vehicles) that lead to better transfer. This work also introduces a novel method for domain adaptation that utilizes adversarial-training and learning with pseudo-labels.
They measure transfer performance across domains (CARLA to NuScenes) on the bird’s-eye-view vehicle segmentation from multi-view images task.

**Limitations And Societal Impact:**

yes

**Main Review:**

This work addresses important questions in domain adaptation for sim2real transfer with limited amounts of labeled data in the target domain.

This paper provides valuable discussion and experiments on how data collection in the source domain should be performed i.e. how does diversity of weather and vehicle assets in the simulator affect downstream transfer? These are important design decisions when developing simulators and I found the experiments on data collection Fig. 7-12 to be well-designed and insightful.

**Questions/Concerns**:
* The authors use only BEV vehicle segmentation for evaluation. I would like to see some discussion on why they chose this over full perception task presented in the Lift-Splat-Shoot (vehicle segmentation, driveable area, lane boundary …).
* I find the ablations on data sampling quite thorough (weather/car color diversity/...) but the current set of experiments provide no insight in how transfer varies across simulator photorealism, a major factor to consider in simulation.
* The proposed training objectives are described in Eq 4, 5, 6 and Figure 14 in the supplementary material. I highly recommend moving Figure 14 to the main paper, as well as including the training objective in the figure.

**Typo/Prose/Figures**:
* Line 245: “deficiencies”
* Line 277: “obtained”
* Line 331: “its”
* Figure 13: images are quite difficult to see


**Time Spent Reviewing:**

5

---

> ### Author Response · Authors · 2021-08-10
> **Thank you for the review**
>
> We thank the reviewer for their valuable feedback. We appreciate that the reviewer believes our paper addresses “important questions”, provides “valuable discussion” on how data collection in the source domain should be performed, and that the reviewer found our experiments “well-designed and insightful”. We respond to the reviewer’s comments below:
>
> 1. “The authors use only BEV vehicle segmentation for evaluation” - the additional tasks tested in “Lift, Splat, Shoot” involve mapping and planning. For mapping tasks, the difficulty is that it is hard to extract map layers like “drivable area” and “lane boundaries” from CARLA in such a way that the conventions match the conventions used to create the map layers in the HD maps provided by nuScenes. For planning, the difficulty is that the CARLA traffic model is very rigid compared to human driving, so there is again a labeling function mismatch that the DA algorithms we test cannot compensate for (theorem 1). We therefore chose to focus only on BEV vehicle segmentation in this work and leave the problem of overcoming the labeling function mismatch for map layers and traffic models to future work.
>
> 2. “the current set of experiments provide no insight in how transfer varies across simulator photorealism, a major factor to consider in simulation” - We are running experiments in which we vary parameters that control the noise models used to render images and point clouds supported by CARLA (https://carla.readthedocs.io/en/latest/ref_sensors/) and will include these in our revisions. We absolutely agree with the reviewer that these results are significant to include from the standpoint of simulation design; although photorealism is heavily sought-after in the graphics community, performance gains are more likely to be bottlenecked by asset diversity and placement and weather simulation than photorealism. This phenomenon was noticed by [1] (section 6) on a different but related task:  “Realism is overrated.... Realistic effects, such as sophisticated lighting models, are not important to learn basic optical flow, but merely induce minor improvements.”
>
> 3. “I highly recommend moving Figure 14 to the main paper, as well as including the training objective in the figure” - We appreciate the feedback. We will move Figure 14 to the main paper in our revisions and edit the figure to include the training objective.
>
>
> References:
>
> [1] Mayer et al “What Makes Good Synthetic Training Data for Learning Disparity and Optical Flow Estimation., IJCV.

---

> > ### Comment · Reviewer_gzio · 2021-08-30
> > **Update**
> >
> > Thank you to the authors for the detailed replies -
> > in light of the additional analysis provided during the rebuttal period I have revised my score.
> >
> > I am still curious to see some updates on (2.) from the previous posts and would like to see if there were any updates on that

---

### Official Review · Reviewer_LRhN · 2021-07-16

**Rating:** 7
**Confidence:** 3

**Summary:**

The paper proposes a method for both generating data in simulation and training with the generated data for the bird's-eye-view (BEV) vehicle segmentation task. It takes inspiration from recent theoretical advances in domain adaptation (DA) and frames learning from simulation as DA. It also points out special properties of using simulated data for adaptation, i.e., we can control the simulated data (label) distribution. Inspired by risk upperbound derived in prior work, it proposes a synthetic data generation strategy that samples data based on human-designed spatial priors and priors of the target domains (if some target labels are given). Once the data is generated, it employs a training method that extends recent work on adversarial DA to the dense prediction task of BEV vehicle segmentation and aims to learn invariant representation across domains. For experiments, the paper synthesizes data using the CARLA simulator and evaluates the adapted model on the nuScenes dataset. Results show that the method outperforms other data synthesis strategies and training methods such as style transfer-based DA.

**Limitations And Societal Impact:**

The paper adequately discussed its limitations and potential negative societal impact. The limitations listed in the papers are mainly typical failure modes of the proposed method, such as mistaking pedestrians for vehicles.

**Main Review:**

**Strength:**

- The idea of optimizing synthetic data distribution to match that of target domain distribution is sensible and appealing. The paper follows a rigorous DA framework to analyze the need for optimizing data sampling distribution in order to minimize the target domain risk.
- Although the theoretical results and DA approach used in the paper are not new, the paper uses them effectively to solve practical real-world tasks such as BEV vehicle segmentation, and shows that the proposed framework outperforms prior data synthesis and DA methods typically used in these tasks. The method also shows good robustness against different types of domain gaps such as maps, vehicle assets, and weather. So this seems like a good empirical contribution.
- The paper is generally well-written. Most techniques are well-motivated from a theoretical standpoint.

**Weakness:**

- When no target label is provided (direct transfer learning), the spatial priors used in Sec 3.1 are rather heuristic, which makes the method less principled or general. Using these human priors is what people typically do when generating synthetic data, so the practice itself is not new, although the paper does formalize it inside a DA framework.
- The paper claims to be "agnostic of the network architecture and the choice of a simulator", however, the paper has only used one simulator, one architecture, and one hand-designed prior. Thus, it is difficult to verify this claim. It would be more interesting if the authors can try other architecture and experiment with other autonomous driving tasks and use other priors.
- In the semi-supervised learning setting, the performance of the proposed method over Lift Splat seems small. So it seems like the method is most effective for direct transfer learning. One could argue that the performance gain is mainly coming from the strong hand-designed spatial priors.

**Additional Comments:**

typo:

- line 245: "defficiencies" → "deficiencies"
- line 331: "it's" → "its"

**Time Spent Reviewing:**

6

---

> ### Author Response · Authors · 2021-08-10
> **Thank you for the feedback**
>
> We thank the reviewer for finding our techniques *“well-motivated from a theoretical standpoint”*, acknowledging that we have used them *“effectively to solve practical real-world tasks”* and finding our synthetic data generation strategy *“sensible and appealing"*.
> We also thank the reviewer for the constructive feedback.
>
> Following your recommendation:
>
> * We have clarified in our paper that we experiment with two architectures: the “Lift, Splat” model for BEV vehicle segmentation from multi-view camera input, and PointPillars for BEV vehicle segmentation from multi-scan LiDAR input.
> * We have emphasized that the reason we only use CARLA in our experimental results is due to the availability of open-source self-driving simulators with well-documented Python API, and not because of a limitation of our method.
> * We have emphasized in the main text that the scope of our work and analysis techniques are focused on the unsupervised scenario (*absence of real world labels* as in line 4-5).
> * We have also emphasized that how to optimally mix data (semi-supervised scenario) is an interesting open-question left to further work.
>
>
>
> Below we address your concerns in more detail.
>
> ---
>
>
>
> > When no target label is provided (direct transfer learning), the spatial priors used in Sec 3.1 are rather heuristic, which makes the method less principled or general. Using these human priors is what people typically do when generating synthetic data, so the practice itself is not new, although the paper does formalize it inside a DA framework.
>
>
>
> **R**: We agree that when no target labels are available the spatial priors could be seen as a heuristic. However, in the absence of data, what else can be done? In these circumstances, we believe the right approach is to use our prior belief of the world. As soon as data becomes available, we can update those priors with real statistics. As pointed out by the reviewer, our paper additionally casts this technique inside the DA framework and motivates why these heuristics improve transfer performance (Theorem 2). We additionally emphasize that while priors could be seen as a heuristic, our intuition is still more principled and grounded than the use of existing techniques (e.g. domain randomization) that increase the divergence between the label marginals.
>
> ---
>
>
>
> > The paper claims to be "agnostic of the network architecture and the choice of a simulator", however, the paper has only used one simulator, one architecture, and one hand-designed prior. Thus, it is difficult to verify this claim. It would be more interesting if the authors can try other architecture and experiment with other autonomous driving tasks and use other priors.
>
>
>
> **R**: In our paper, we use two architectures, one for each task: “Lift, Splat” for BEV vehicle segmentation from multi-view camera input and PointPillars for LiDAR input. We also compare different spatial priors. For example, for our data generation baseline (Road Structure), we sampled data according to the structure of the road. This method is inspired by [48, 34] and it effectively defines a baseline prior (line 264). The experiments on the robustness to town also implicitly define different priors ( e.g. as the topology of the road changes).
>
> **Simulator:**  As mentioned above, we will clarify in our revisions that we use CARLA because it is an open-source driving simulator that is well-used in the driving community, not due to a limitation of our method.
>
> ---
>
>
>
> > In the semi-supervised learning setting, the performance of the proposed method over Lift Splat seems small. So it seems like the method is most effective for direct transfer learning. One could argue that the performance gain is mainly coming from the strong hand-designed spatial priors.
>
> We agree that as more labeled real data becomes available, the performance saturates. Similar observations have been obtained in the domain-adaptation literature (e.g. see section 5 and section 6 in [3]). There are many plausible reasons for explaining this trend and we plan to investigate them more in the future as data mixing is an important problem setting for the self-driving community. However, we believe this investigation may require additional mathematical machinery and we defer it to further work. Our goal in this paper, as stated in our abstract, is to motivate strategies for using a self-driving simulator in the *absence of real world labels* -- a simpler initial use case. Our goal with the semi-supervised experiment is to demonstrate that our methods additionally achieve gains when some real data is used. We believe this result showcases the generality of our method.  We thank the reviewer for this question, and will add further clarification about our problem setting in the paper.

---

> > ### Comment · Reviewer_LRhN · 2021-08-31
> > **Thank you for the response**
> >
> > Thanks to the authors for the response, which has largely addressed my concerns, therefore I have upgraded my score.

---

### Official Review · Reviewer_z4FW · 2021-07-17

**Rating:** 7
**Confidence:** 3

**Summary:**

The authors propose a technique for sampling data from a self-driving car simulator. They sample in a simulator-agnostic way in an effort to reduce the distance between the label marginals and therefore reduce the sim2real gap.They compare their approach to standard domain adapation techniques. They compare their techniques to standard data adaption frameworks and baselines and show promising results. They also demonstrate that domain adversarial training outperform simple style transfer approaches.

**Limitations And Societal Impact:**

Yes

**Main Review:**

# Originality:
The paper follows in a long line of adapting self driving car simulator data to the real world. The individual ideas in the paper are different from the related works mentioned since it relies on regenerating the datasets using parameters from the simulator. In this way, it's slightly different from DANN or MUNIT. The lack of related work that is actually compared to in the paper is my biggest concern. Note: not many papers focus on what parts of the simulator are actually relevant to transfer between datasets or the limitations of the simulator itself and this is is a very valuable and understudied problem.

# Quality
The paper does appear to include  proper comparisons against the other techniques. The claims are well supported technically (with the exception of the technique being simulator agnostic. That particular point is not demonstrated by the paper.

Truly showing that would require more simulators than just CARLA, but that is a minor issue.  The baselines compared against seem adequate. The weakest portion of the paper is the limited number of baselines. They only compare to DANN or MUNIT despite those baselines being rather old. More modern state of the art domain adaption and style transfer strategies. DANN was published 5 years ago in 2016.  Without recent and adequate baselines, the paper feels incomplete. The fact that only rather dated two baselines were chosen is rather concerning as reviewer as it seems possible that more recent work may outperform this approach. The shear lack of baselines here is my greatest concern for this paper.

# Clarity
The clarity of the paper is adequate. Note that many of the figures are so small that they are hard to read the text on, but this  is easily fixable. (In fact a lot of the figures are small enough that they are bit hard to read the underlying text). The paper is well written.

# Significance
This paper would prove helpful in the domain adaption self-driving cars. As self-driving car research requires accurate simulators that transfer well, this submission would prove useful in that space. More importantly though, this paper is very relevant to demonstrating which types of variation (ie. map design, and number of vehicles) actually matter to self-driving car simulator allowing the creators to focus on the types of variation that are most important to impactful real world performance. For instance, they state that variation in NPC color does not matter for performance so there is no need to build a randomization strategy for those NPC colors. On the other hand, they showing matching the NPC positions with a dataset like nuscene is rather important.

#Update
Given the rebuttals from the authors, I am willing to raise my score to a 7. However, I would recommend the do not include their claims about the method being simulator agnostic as while possible, this claim has not be empirically proven in the paper through the use of multiple simulators.

**Time Spent Reviewing:**

4

---

> ### Author Response · Authors · 2021-08-10
> **Thank you for the feedback**
>
> We thank the reviewer for finding that our work is *“well supported technically”*, *“very relevant to demonstrating which types of variation actually matter to self-driving car simulator”* and includes *“proper comparisons against the other techniques”*.  We also thank the reviewer for the constructive feedback.
>
> Following your recommendation:
>
> * We have emphasized in our paper that the reason we only used CARLA in our experimental results was due to the availability of complete open-source self-driving simulators with a well-documented Python API, and not because of a limitation of our method.
> * We will add a new baseline inspired by [1] (winner of the Domain Adaptation track of the 2020 Waymo Open Dataset Challenge) to Table 1/3.
>
>
>
> Please find detailed answers to your questions below:
>
> ---
> > The technique being simulator agnostic. That particular point is not demonstrated by the paper. Truly showing that would require more simulators than just CARLA, but that is a minor issue.
>
> As mentioned above, we will clarify in our revisions that we use CARLA because it is an open-source driving simulator that is well-used in the driving community, not due to a limitation of our method.
>
> ---
> > The weakest portion of the paper is the limited number of baselines. They only compare to DANN or MUNIT despite those baselines being rather old. More modern state of the art domain adaption and style transfer strategies.
>
> We answer this question in two parts, as the contribution of our work is two-fold.
>
> **(1)** *Baseline for the Synthetic Data Generation technique*. In this case, we proposed a challenging baseline, which we called road structure (RS). This baseline is inspired by [48,34] but within the realistic conditions of the self-driving simulator. If there are specific suggestions for baselines the reviewer would like us to add, we are happy to include them as well.
>
> **(2)** *Baseline for the Training Strategy*. We emphasize that the framework proposed in DANN is still the cornerstone of many state-of-the-art complex CV sim2real tasks as many domain-adaptation methods cannot be simply extended to complex computer vision tasks. If there are other specific baselines the reviewer believes would be valuable to include, we are happy to include these as well. Please let us know.
>
> ---
> > Many of the figures are so small that they are hard to read the text on, but this is easily fixable.
>
> We appreciate the feedback and will make our figures bigger in the revision.
>
>
> References:
>
> [1] 1st Place Solution for Waymo Open Dataset Challenge - 3D Detection and Domain Adaptation https://arxiv.org/pdf/2006.15505.pdf

---

> > ### Author Response · Authors · 2021-08-26
> > **Gentle Reminder**
> >
> > Dear Reviewer z4FW,
> >
> > Thank you very much again for the time and effort put into reviewing our paper. We believe that we have addressed all your concerns in our response. We have also followed your suggestion to improve our paper and have added additional experimental results ( see general response ). We kindly remind you that we have less than a week available for the discussion period. We would love to know if there is any further concern, additional experiments, suggestions, or feedback, as we hope to have a chance to reply before the discussion phase ends.

---

### Official Review · Reviewer_TDjR · 2021-07-18

**Rating:** 5
**Confidence:** 5

**Summary:**

This paper tackles the problem of sim2real domain adaptation for the bird's-eye-view vehicle segmentation task. The authors trained existing models (Lift Splat and Point Pillar) with carefully sampled synthetic data,  pseudo-labels on real data (eq. 6), and gradient-reversal layers with f-DAL inspired loss. On the nuScenes dataset, the authors found that the these tricks improve models' (Lift Splat and Point Pillar) sim2real generalization accuracy of the vehicle segmentation IOU (Table 1 and Table 2).

**Ethical Concerns:**

There is no ethical concerns

**Limitations And Societal Impact:**

Yes (see: section 7)

**Main Review:**

Strengths:
- The authors study a few recent sim2real techniques (e.g., pseudo-labels on real data, gradient-reversal layers with f-DAL inspired loss).
- The authors carefully sampled synthetic data from simulators to match the real labels marginals of the nuScenes datasets (section 3).
- Using the above tricks, two models' (Lift Splat and Point Pillar) sim2real generalization accuracies improve on the vehicle segmentation IOU metrics (Table 1 and Table 2).

Weaknesses:
- The paper combines a few existing techniques to train baseline models on synthetic data. The techniques and baseline models are all from existing literatures, so the novelty is lacking.
- The authors didn't include comparison with prior works on the nuScenes datasets. What are the current SOTA models' accuracies (trained on real data or sim-real mixed)? Are there other sim2real works that evaluate on nuScenes?

**Time Spent Reviewing:**

4

---

> ### Author Response · Authors · 2021-08-10
> **Thank you for the review**
>
> We thank the reviewer for their valuable feedback. We address the reviewer’s comments below.
>
> 1. "The techniques and baseline models are all from existing literatures, so the novelty is lacking." - Our goal in this paper is to identify effective strategies for 1) sampling training data from a graphics simulator, and 2) training on this synthetic data, such that models perform well when tested on real-world data. While we do utilize existing theoretical results from the literature, it is how we utilize them that leads us to develop a practical method for training in simulation; as pointed out by Reviewer-​​LRhN, our work “uses [DA] effectively to solve practical real-world tasks such as BEV vehicle segmentation”.
> We believe that our domain adaptation formalism (section 2) that guides how best to generate synthetic data for a multi-sensor perception stack for self-driving (section 3), along with our empirical study (section 6) of what simulation-reality gaps can and cannot be effectively compensated for by better domain adaptation algorithms, are valuable and novel contributions to the self-driving community. We will update the language in our “Contributions” subsection to clarify our claims about novelty.
>
> 2. “The authors didn't include comparison with prior works on the nuScenes datasets.” In the Domain Adaptation track of the 2020 Waymo Open Dataset Challenge (https://arxiv.org/pdf/2006.15505.pdf), the top performing model used ensembling and test-time augmentation. We are happy to add an analogous baseline using ensembling and test-time augmentation to our paper (Table 1/2). If there are other specific baselines the reviewer believes would be valuable to include, we are happy to include these as well.
>
> 3. “Are there other sim2real works that evaluate on nuScenes?” Prior work on domain adaptation for self-driving has primarily focused on single-image semantic segmentation and tested on Cityscapes or BDD100K. Since one of our main goals is to test how different aspects of simulation affect both multi-view camera and LiDAR tasks, we chose to test on nuScenes as nuScenes is one of the largest datasets including both LiDAR and a 360-degree camera rig with precise calibration. We believe that the release of the datasets we generated with CARLA that mimic nuScenes will stimulate future work on techniques for training any-modal self-driving perception models on synthetic data.

---

> > ### Author Response · Authors · 2021-08-26
> > **Gentle Reminder**
> >
> > Dear Reviewer TDjR,
> >
> > Thank you very much again for the time and effort put into reviewing our paper. We believe that we have addressed all your concerns in our response. We have also followed your suggestion to improve our paper and have added additional experimental results ( see general response ). We kindly remind you that we have less than a week available for the discussion period. We would love to know if there is any further concern, additional experiments, suggestions, or feedback, as we hope to have a chance to reply before the discussion phase ends.

---

### Author Response · Authors · 2021-08-26
**General Response and Additional Baselines**

We again thank the reviewers for their thoughtful comments on our work. We are very grateful for the comments provided. e.g.  **(*R-z4FW*)**  *“the claims are well supported technically”* , *“ this paper is very relevant to demonstrating which types of variation actually matter to self-driving car simulator”*  ;   **(*R-LRhN*)**  *“Most techniques are well-motivated from a theoretical standpoint ”*, *“this paper uses them [DA results] effectively to solve practical real-world tasks”*;   **(R-gzio)**  *“This work addresses important questions”*, *“this paper provides valuable discussion and experiments on how data collection in the source domain should be performed”*, *These are important design decisions when developing simulators and I found the experiments on data collection Fig. 7-12 to be well-designed and insightful*.

---

Following the additional recommendations from  **R-TDjR** and **R-z4FW**, we have added new baselines inspired by [2] (winner of the Domain Adaptation track of the 2020 Waymo Open Dataset Challenge) to Table 1. *Our method outperforms these baselines by a noticeable amount as shown below*.

| Method   | Performance |
| :------- | ----------- |
| RS-No Adaptation | 9.76  |
| *RS-Ensemble (Inspired by [2]) (NEW)* | 11.95 |
| RS-Style Transfer | 13.33 |
| RS-DANN | 13.76 |
| *RS-Ensemble + Test-time Augmentation (Inspired by [2]) (NEW)* | 13.76|
| Ours | **17.84**  |

---

**Explanation of the new baselines:** For the ensemble baselines, we train 4 models on CARLA-generated data. These models individually achieve nuScenes transfer IOUs of {9.749, 9.877, 8.726, 9.044}. For the *RS-Ensemble* baseline, we make a prediction with each model and take the prediction with the highest IOU with respect to ground-truth. For the *RS-Ensemble + Test-time Augmentation* baseline, we randomly transform the input images according to the same augmentation parameters used in “Lift Splat” [1], 8 times for each of the 4 models and take the prediction with the highest IOU with respect to ground-truth. *Note that due to the fact that these ensemble baselines use ground-truth to choose the prediction, they represent an upper-bound on the performance of ensemble baselines used in the Waymo adaptation challenge* [2].

References:

[1] https://arxiv.org/abs/2008.05711

[2] https://arxiv.org/abs/2006.15505

---

### Decision · Program_Chairs · 2021-09-27

**Decision:**

Accept (Poster)

**Comment:**

This paper studies the problem of sim2real domain adaptation, focusing on the bird's-eye-view vehicle segmentation task and on strategies for sampling data from a self-driving car simulator. The authors trained existing models and sampled data in a simulator-agnostic way in an effort to reduce the sim2real gap, compared their approach to standard domain adaptation techniques and demonstrated that domain adversarial training outperformed simple style transfer approaches.

Reviewers have praised the clarity of the paper and thoroughness of the experiments, and noted that this research was “very relevant to demonstrating which types of variation (ie. map design, and number of vehicles) actually matter to self-driving car simulators”.
Reviewers’ claims about “novelty” are easy to make and to dismiss, and I note that the authors provide an in-depth mathematical formalism and analysis of the simulation-reality gap.
Reviewers also had concerns about missing evaluation and comparison of SOTA models on nuScenes (TDjR and z4FW) - the authors have addressed these by running additional experiments comparing their method with “HorizonLiDAR3D" from the 3D detection track and the domain adaptation track in Waymo Open Dataset Challenge at CVPR 2020, training the models on CARLA; moreover reviewer z4FW praised the “focus on what parts of the simulator are actually relevant to transfer between datasets”. Reviewers gzio and LRhN had additional small concerns about photorealism, focus on bird eye view, heuristics and spatial priors, and the authors committed to answer them.

Reviewers TDjR and z4FW did not answer the authors’ rebuttal but I believe they should and would increase their score. I am therefore willing to challenge these two reviewers and promote this paper to an acceptance.